# Operative Re-Intervention following Pancreatoduodenectomy: What Has Changed over the Last Decades

**DOI:** 10.3390/jcm11247512

**Published:** 2022-12-19

**Authors:** Jana Enderes, Christiane Pillny, Jens Standop, Steffen Manekeller, Jörg C. Kalff, Tim R. Glowka

**Affiliations:** 1Department of Surgery, University Hospital Bonn, 53127 Bonn, Germany; 2Department of Surgery, DRK-Hospital Neuwied, 56564 Neuwied, Germany

**Keywords:** operative reintervention, redo surgery, reoperation, pancreatoduodenectomy, whipple, pancreatic fistula

## Abstract

Background: To investigate changes over the last decades in the management of postoperative complications following pancreatoduodenectomy (PD) with special emphasis on reoperations, their indications, and outcomes. Methods: 409 patients who underwent PD between 2008 and 2021 were retrospectively analyzed with respect to their need for reoperations (reoperation, n = 81, 19.8% vs. no reoperation, n = 328, 80.2%). The cohort was then compared to a second cohort comprising patients who underwent PD between 1989 and 2007 (n = 285). Results: 81 patients (19.8%) underwent reoperation. The main cause of reoperation was the dehiscence of pancreatogastrostomy (22.2%). Reoperation was associated with a longer duration of the index operation, more blood loss, and more erythrocyte concentrates being transfused. Patients who underwent reoperation showed more postoperative complications and a higher mortality rate (25% vs. 2%, *p* < 0.001). Compared to the earlier cohort, the observed increase in reoperations did not lead to increased mortality (5% vs. 6%, *p* = 353). Conclusions: The main cause for reoperation has changed over the last decades and was the dehiscence of pancreatogastrostomy. Associated with a leakage of pancreatic fluid and clinically relevant PF, it remains the most devastating complication following PD. Strategies for prevention and treatment, e.g., by endoscopic vacuum-assisted-closure therapy are of utmost importance.

## 1. Introduction

Pancreatic surgery, specifically pancreatoduodenectomy (PD), is a very demanding procedure and requires the technical skills of experienced and specialized surgeons with high individual caseloads. Mortality after PD in Germany is low at 6.1% at high-volume centers [1] and even lower in the United States at 2.7% [2]. Even if it is carried out with the greatest care, PD is still known to be associated with high morbidity, ranging from 30 to 50% [3]. The specific complications after PD are postpancreatectomy hemorrhage (PPH) in 1–8% [4], pancreatic fistula (PF) in 3–45% [5], and delayed gastric emptying (DGE), which is observed in up to 60% of cases [3]. Common general postoperative complications, such as wound infections and postoperative ileus, but also non-surgical complications such as pneumonia and thrombosis, are also observed after PD [6]. Most complications can be treated without any intervention; however, some complications need interventional treatment strategies, such as CT-guided or endoscopic interventions, and operative re-interventions are frequently observed after PD [7,8,9,10,11].

In the past, we analyzed the outcome of operative re-interventions conducted between 1989 and 2007 at our center following PD specifically in regard to secondary surgery [7]. The main reasons for reoperation were intra-abdominal bleedings, infectious fluid collections, and complications at the laparotomy side with a concomitant increase in the length of the hospital stay. With the introduction of the standardized definitions of the International Study Group of Pancreatic Surgery (ISGPS) in 2008, the purpose of the present study was to investigate what has changed after the implementation of these definitions over the last decades between 2008 and 2021 with respect to the management of postoperative complications following pancreatoduodenectomy with a special emphasis on operative re-interventions, their indications, and outcomes.

## 2. Materials and Methods

Between 2008 and 2021 a total of 411 patients underwent PD at our center. All of these patients gave their written informed consent to prospectively record their data in a pancreatic resection database, which was also approved by the institutional ethics committee (ethics committee of the Rheinische Friedrich-Wilhelms University Bonn, 347/13). Excluded were patients who underwent PD at other hospitals and were specifically referred to us for secondary surgery due to major complications (n = 2). Thus, included were 409 patients who were retrospectively analyzed regarding their need for operative re-intervention (reoperation, n = 81, 19.8% vs. no reoperation, n = 328, 80.2%), their indications, and outcomes in relation to perioperative factors. Postoperative complications in general were documented according to the Clavien-Dindo classification [12] and the specific complications associated with pancreatic surgery such as PF, PPH, and DGE were classified according to the definitions of the International Study Group on Pancreatic Surgery [3,4,5]. The cohort was then compared to a second cohort comprising patients who underwent PD between 1989 and 2007 (n = 285) [7] in order to assess what has changed over the last decades with regard to the outcome after PD with special emphasis on operative re-interventions.

PD was performed by five certified senior pancreatic surgeons (JCK, SM, JS, NS, TRG). Surgery was carried out in a standardized way with single-loop reconstruction, either ante- or retrocolically and if the latter was chosen, supra- or infracolic routes were used [13]. Double-loop reconstruction was only carried out in case of infiltration of the antrum and necessary Whipple procedure. Duodenoenterostomy, pancreatogastrostomy by default, and end-to-side choledochojejunostomy were carried out as previously described [14,15]. Perioperatively, every patient received an epidural catheter for postoperative analgesia, a nasogastric tube (NGT), and two soft drains at the sites of pancreatogastrostomy and choledochojejunostomy before closure of the abdomen. In case of contraindications for an epidural catheter, opioids were given via patient-controlled analgesia. In case of preoperative cholangitis, patients were treated with antibiotics. Preoperative stenting was only performed if a surgery-first approach was not possible, e.g., in case of neoadjuvant chemotherapy. Perioperative care was carried out according to our institutional standard enhanced recovery after surgery (ERAS) protocol as follows: in case of severe weight loss, patients received additional enteral sip feeds or parenteral nutrition prior to surgery. Patients did not receive bowel preparations and were allowed solid food and liquids 6 and 2 h before surgery, respectively. Directly after surgery patients were allowed to drink water or tea, and only if amylase levels were normal on postoperative day (POD) 3, patients were allowed an easily digestible/fat-reduced diet followed by an easily digestible/fiber-reduced diet on POD4, a basic diet (no pulses/no brassica) on POD5 and a normal diet on POD 6. In the case of vomiting, transition to a normal diet was discontinued and an NGT was re-inserted. The intraoperatively administered NGT was removed if daily secretions were less than 500 mL, and soft drains were removed if amylase levels on POD 3 were normal. In case of development of pancreatic fistula, measured by amylase levels in abdominal drains on POD 3 by default, octreotide (100 μg 3x/d s.c.) was given for five days.

Statistical analysis was conducted using Excel 2013 (Microsoft Corporation, Redmond, Washington, DC, USA) and SPSS 28 (IBM Corporation, Armonk, NY, USA). Non-normally distributed variables data were expressed as medians and interquartile range and analyzed using the Mann–Whitney U test. Categorical data were expressed as proportions and compared with the Pearson x^2^ or the Fisher’s exact test. In the univariate analysis variables with *p* < 0.1 were included in multivariate stepwise logistic regression with a significance level of *p* < 0.05 for inclusion and *p*< 0.10 for removal in each step. The relative risk was described by the estimated odds ratio with 95% confidence intervals. A *p*-value <0.05 was considered statistically significant.

## 3. Results

Of the 409 patients included in the study and undergoing PD, 187 (45.7%) were for pancreatic adenocarcinoma, followed by chronic pancreatitis (49 patients, 12.0%), malignant papillary tumors (39 patients, 9.5%), and distal bile duct cancer (39 patients, 9.5%) (Figure 1). There were 160 (39%) female and 249 (61%) male patients with an average age of 66 (±12) years.

Out of the 409 patients, 81 patients (19.8%) underwent operative re-intervention, whereas the majority of patients (328 patients, 80.2%) did not need a second surgery. Patients undergoing operative re-intervention were mainly male (70% male vs. 30% female), showed an average age of 67 (58–73) years, and had a BMI of 25.1 (22.3–28.7) kg/m^2^ (Table 1). Gender, age, as well as the height and weight measured by BMI did not differ between patients who underwent operative re-intervention and patients without a need for a second surgery. However, patients who underwent operative re-intervention showed a higher CCI (2 (1–3) vs. 2 (2–3), *p* = 0.046). Preoperative biliary drainage occurred equally in both groups (42% vs. 51%, *p* = 0.193), even though cholangitis was more often observed in the operative re-intervention group (30% vs. 16%, *p* = 0.005) (Table 1).

Operative re-intervention was associated with a longer duration of the index operation (458 (355–556) min vs. 401 (333–473) min, *p* = 0.003), more blood loss (800 (475–1500) ml vs. 600 (375–1000) mL, *p* = 0.036), and more erythrocyte concentrates being transfused (1 (0–4) vs. 0 (0–2), *p* = 0.001) during the index operation (Table 2). Patients who underwent a second surgery showed—at least by trend—a softer pancreas parenchyma compared to patients without a need for reoperation.

The indications for operative re-intervention were dehiscence of the pancreatogastrostomy (18 patients, 22.2%) followed by PPH (15 patients, 18.5%)—of which 11 were intraabdominal (extraluminal) (13.6%) and 4 were bleedings at pancreatogastrostomy site (intraluminal) (4.9%), surgical site infections (15 patients, 18.5%), and insufficiency of BDA/bile leakage (11 patients, 13.6%) (Figure 2). In total, 3 patients (3.7%) had to undergo a second surgery due to infectious fluid collections.

The patients who underwent reoperation showed more postoperative complications, such as a higher rate of insufficiency of BDA (17% vs. 3%, *p* < 0.001) and DE (35% vs. 1%, *p* < 0.001), more suprafascial wound infections (48% vs. 16%, *p* < 0.001), and intraabdominal abscess formations (31% vs. 11%, *p* < 0.001), as well as more PF grade B/C 44% vs. 14%, *p* < 0.001), PPH grade B/C (37% vs. 22%), *p* = 0.002), and DGE grade B/C (30% vs. 19%, *p* = 0.002) (Table 3). These postoperative complications not only led to a longer stay in the ICU (6 (2–16) d vs. 2 (1–3) d, *p* < 0.001) but also to a longer postoperative stay in general (38 (22–64) d vs. 20 (15–26) d, *p* < 0.001). In addition, patients undergoing reoperation showed a higher mortality rate (25% vs. 2%, *p* < 0.001) (Table 3).

Risk factors associated with high mortality are shown in Table 4. In the univariate analysis, reoperation, PF grade B/C, PPH grade B/C, intraabdominal abscess formation, and positive intraoperative microbiology qualified for multivariate analysis, in which only reoperation indeed seemed to be a risk factor for a higher mortality rate.

The results of the comparison between patients who underwent PD at our center between 1989 and 2007 (n = 285) [7] and patients who underwent PD from 2008 to 2021 (n = 409) are shown in Table 5. Within the latter time period, patients were older, with 9% showing an average age > 80 years compared to 3% in the former period (*p* = 0.002). The ASA classification did not differ between the two cohorts. Intraoperatively, within the latter period, fewer erythrocyte concentrates were being transfused (2 (0–4) vs. 0 (0–2), *p* < 0.001), and—at least by trend—we observed less blood (800 (400–1300) mL vs. 600 (400–1000) mL, *p* = 0.075)**.** As for postoperative complications according to ISGPS, we detected more PF, PPH, and DGE and were also able to observe more reoperations (11% vs. 20%, *p* = 0.002), which presumably both led to a significantly longer postoperative stay even though the duration of the ICU stay was not prolonged but even shortened in the latter period. Interestingly and far most important, the mortality rate did not differ between the two periods (5% vs. 6%, *p* = 0.353).

## 4. Discussion

PD is still known to be associated with high morbidity, ranging from 30 to 50% [3]. Since most complications can be treated without any intervention, some need interventional treatment strategies, and others even require operative re-interventions. In our cohort reoperation was necessary in 20% of the cases, which is in accordance with other studies reporting a reoperation rate from 5 to 18.5% [8,9,10,11]. The rate of secondary surgery at our center seems to be placed rather at the higher end of this range; however, comparisons between these studies have to be drawn carefully since study design and exclusion criteria differ between these studies. The two studies reporting a reoperation rate as low as 5 and 6.7% [9,10], for instance, did not include emergency pancreatic resections, which are known to be associated with a higher reoperation rate [16]. As suspected, we observed more postoperative complications amongst patients who underwent reoperation, which subsequently led to a longer postoperative stay and a dramatic increase in mortality as seen in other studies [8,9,11].

Patients who underwent reoperation showed more comorbidities measured by a higher CCI score. Comorbidities lead to a higher need for reoperations as was observed at least for other major abdominal surgeries [17]. When comparing other studies focusing on reoperations after PD, comorbidities are described to have an impact on the rate of reoperation. Chronic obstructive pulmonary disease was more frequently observed amongst patients who underwent reoperation [9,11]. Moreover, diabetes mellitus was a risk factor in multivariate analyses [9,10]. In the former study by Lyu et al., in addition, an ASA score of three and four was considered a risk factor for reoperation in the multivariate analysis; however, in the same study preoperative functional status was not. Qiu et al. did not find a difference in the ASA scores even though, as mentioned, comorbidities, such as diabetes mellitus, were considered risk factors in the multivariate analysis [10]. Due to these conflicting results regarding the ASA score and actual comorbidities, we believe that CCI is more suitable to assess comorbidities than the ASA score is. Interestingly, we observed preoperative cholangitis more frequently in patients undergoing reoperation, even though preoperative biliary drainage was distributed equally amongst the two groups, reflecting a more progressive surgery-first approach in patients with preoperative cholangitis, as suggested in a meta-analysis comprising 22 retrospective and 3 randomized controlled trials [18]. Other studies also reported elevated preoperative bilirubin levels, as a possible indicator for cholangitis, in the reoperation group with equally distributed preoperative biliary drainage [8,9,11]. Preoperative cholangitis was found to be an independent risk factor for mortality after PD [19]. Since our patients in the reoperation group more frequently showed preoperative cholangitis, this could account for the high mortality observed in this group—at least to some extent. However, the results have to be interpreted carefully and cannot be transferred to our study to their full extent since the study by Darnell et al. only included patients who pursued neoadjuvant chemotherapy followed by resection. In our study, only 16 patients, out of the 409 patients included, received chemotherapy before surgery. Interestingly, preoperative cholangitis did not have an impact on postoperative complications in the mentioned study.

Patients who underwent reoperation had a longer duration of the index operation, more blood loss, and more erythrocyte concentrates were being transfused amongst them. Comparing the above-mentioned studies that focused on reoperation after PD, the study by Reddy et al. uncovered blood loss and administration of blood transfusions as risk factors for reoperation in the univariate analysis [8]. Qui et al. also reported more intraoperative blood loss and subsequently more blood transfusions being administered during the index operation in patients undergoing secondary surgery to a later time point and found blood loss over 400 mL in the index operation an independent risk factor for reoperation [8,10]. Lyu et al. did not analyze blood loss or the need for blood transfusions, however, they, as well as Redy et al., reported a longer duration of the index operation in the reoperation group [8,9] whereas Qui et al. did not report any difference. Not only are intraoperative blood loss, subsequent transfusions of erythrocyte concentrates, and a longer duration of the operating procedure associated with a worse outcome after PD in general [20,21], but they also seem to be associated with reoperation to a certain extent. For instance, only just recently intraoperative blood loss >700 mL, as observed in our reoperation group, was reported to be an independent risk factor for the development of clinically relevant PF [22].

Interestingly, indications for redo surgery after PD at our center have changed over the last decades. Whereas intraabdominal bleeding, infectious fluid collections, and complications at the laparotomy side were the main reasons for redo surgery between 1989 and 2007, the main indication for operative re-intervention between 2008 and 2021 was dehiscence of PG, followed by PPH and surgical-site infections. Dehiscence of PG is associated with subsequent leakage of pancreatic fluid into the abdominal cavity, also known as pancreatic fistula (PF), and PFs requiring reoperation are referred to as grade C PF [5]. Importantly, not every patient with dehiscence of PG required surgery, and if possible an interventional approach, which is associated with a better outcome [23], was chosen. Only in case of additional extraluminal hemodynamic relevant bleedings and septic conditions, a second surgery was required. Out of the 15 patients who underwent operative reintervention for dehiscence of PG, 7 required completion of pancreatectomy, in 4 cases, the pancreatic remnant was reinserted, and in another four cases, it was left in situ followed by pancreatic duct occlusion. Risk factors for PF with underlying dehiscence of PG and specific scores to predict PF have been developed over the last years [24,25,26]. The original pancreatic fistula risk score (o-FRS) comprises the following four factors: extensive intraoperative blood loss, small pancreatic duct diameter, soft pancreatic parenchyma, and certain pathologies [24]. The score has been validated and used widely [27], however it has been recently modified with the alternative fistula risk score (a-FRS). This simplified score only comprises soft pancreatic texture, a small duct diameter, and a high BMI [25]. Just recently, in an updated version (updated alternative fistula risk score; ua-FRS) male sex was added as a risk factor and also comprised minimal invasive procedures [26]. In our study, neither small pancreatic duct size, nor BMI, nor male sex were significantly observed more often in the reoperation group, however, pancreatic gland tissue was softer in the reoperation group at least by trend. The experience of the surgeon performing the pancreatic anastomosis is another factor influencing the frequency of PG dehiscence in our opinion. The five certified surgeons performing the surgeries in our cohort did not differ significantly in terms of experience (measured by lifetime and 5-year PDs performed). Moreover, the type of anastomosis (pancreatogastrostomy by default at our center) does not influence the rate of PF according to randomized controlled studies [28]. When comparing the indications for redo surgery throughout the literature Lessing et al. reported anastomotic leakage as the main indication for reoperation with 37.5% including leakage at gastrojejunostomy, pancreaticojejunostomy, and hepaticojejunostomy followed by PPH with 25% [11], which is similar to our study. Reddy et al. and Qiu et al. reported PPH as the main indication for reoperation with 68% and 54.5%, respectively, followed by pancreaticoanastomotic leak with 13% and 22.7%, respectively [8,10].

The observed shift in indications over the last decades could—at least to a certain extent—reflect the progress in radiological techniques. Radiologic interventional treatment possibilities have gained importance in the management of surgical complications after PD [29], also, at our center, infectious fluid collections have increasingly been treated by radiologic CT-guided drainage, and interventional angiography has also become an alternative to treat intraabdominal bleedings [30]. Subsequently, this led to fewer reoperations and a shift in the indication for secondary surgery.

Aside from the detected shift in indications in this study, we in fact performed more reoperations after PD during the second period between 2008 and 2021 (reoperation rate 20%) compared to PDs that were performed during the first period between 1989 and 2007 (reoperation rate 11%). The reoperation rate in the above-mentioned studies ranged from 5% up to 18.5% [8,9,10,11]. Due to the generally higher life expectancy, patients undergoing PD have become older over the years, a phenomenon that we as well observe with a significantly higher percentage of patients over the age of 80 years. High age, specifically an age of 80 years or older, is associated with a higher frequency of comorbidities, also after PD [31,32,33]. These so-called octogenarians automatically have more comorbidities, which one would assume leads to a higher need for reoperations as it was observed at least for other major abdominal surgeries [17] as stated above. However, in a large meta-analysis by Kim et al. octogenarians did not show a higher rate of reoperation [33]. At our center documentation of comorbidities by assessing CCI only started in 2008 and access to patients’ analog files is limited, which is why we cannot draw quantitative comparisons between the two periods here. During the second period, fewer erythrocyte concentrates were being transfused reflecting the adherence to current transfusion guidelines that have become more restrictive over the last decade since there was no statistically significant difference in blood loss between the two periods. Moreover, blood management has become more standardized during the last decade at our center, including preoperative risk stratification and optimization e.g., by treating anemia and iron deficiency and intraoperative careful dissection and the standardized use of ligature devices. As for postoperative complications according to ISGPS, we observed more PPH, PF, and DGE during the second period. Whether this is due to a higher awareness of documentation of the specific postpancreatectomy complications due to the introduction of the ISGPS definitions cannot be asserted with absolute certainty, but it is highly likely. Reasons for a truly increased rate of PPH could include the fact that more borderline resectable tumors in close proximity to arterial vessels are now considered to be operable tumors [34] and also dual platelet inhibition is no longer a contraindication for major abdominal surgery [35] with both comprising a consecutive risk of postoperative bleeding. PPH at least did not lead to more operative re-interventions due to intraabdominal bleeding or bleeding at the gastrostomy site as the switch in indications for secondary surgery shows. Consecutively, due to the higher morbidity observed during the second period, patients had a longer hospital stay compared to the first period. Interestingly this did not include a longer stay in the ICU—on the contrary, the stay in the ICU was even shorter—and far most important this did not lead to an increase in mortality.

This study reveals a shift in the indication for operative re-intervention with dehiscence of PG as the main cause for redo surgery. Associated with leakage of pancreatic fluid and clinically relevant PF, it remains the most devastating complication following PD and thus strategies for prevention and treatment are of utmost importance. We have previously published a multidisciplinary case study of eight patients with PF due to underlying dehiscence of PG [36]. These patients with PF grade B/C with the abdominal fluid collection were either treated with transgastric or transcutaneous CT-guided drainage. In the case of endoscopically diagnosed dehiscence of PG (and without extraluminal hemodynamic relevant bleedings or septic conditions) an endoscopic vacuum-assisted closure therapy (EVT) was initiated. EVT led to a complete closure of the dehiscence in seven cases, one patient died before EVT was finished due to hemorrhagic shock. Most importantly, once initiated EVT did not require further operative interventions. Thus, if initiated early enough, EVT might be a promising interventional strategy to prevent the development of dehiscence that is associated with extraluminal hemodynamic relevant bleedings or septic conditions requiring further surgery. However, to draw valid conclusions, especially with respect to the determination of the optimal time point at which to initiate EVT or the question of which patients might benefit from EVT, further studies investigating EVT in detail need to follow.

## 5. Conclusions

Operative re-interventions after PD are still associated with a high mortality rate. The main cause for reoperation has changed over the last decades and was the dehiscence of PG. If initiated early enough, possible interdisciplinary treatment with endoscopic vacuum-assisted closure therapy might help to prevent the development of a dehiscence that requires surgical intervention and thus might help to further prevent a second surgery resulting in a reduced postoperative mortality.

## Figures and Tables

**Figure 1 jcm-11-07512-f001:**
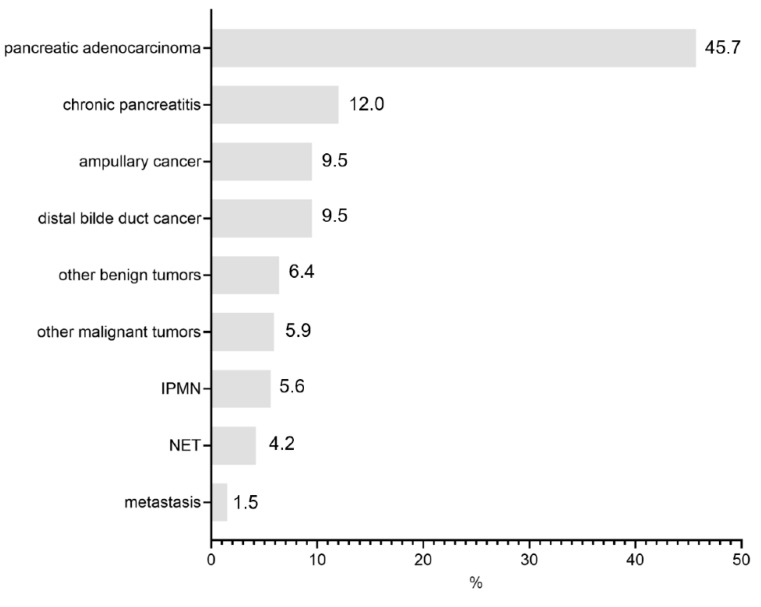
Indications for index operation.

**Figure 2 jcm-11-07512-f002:**
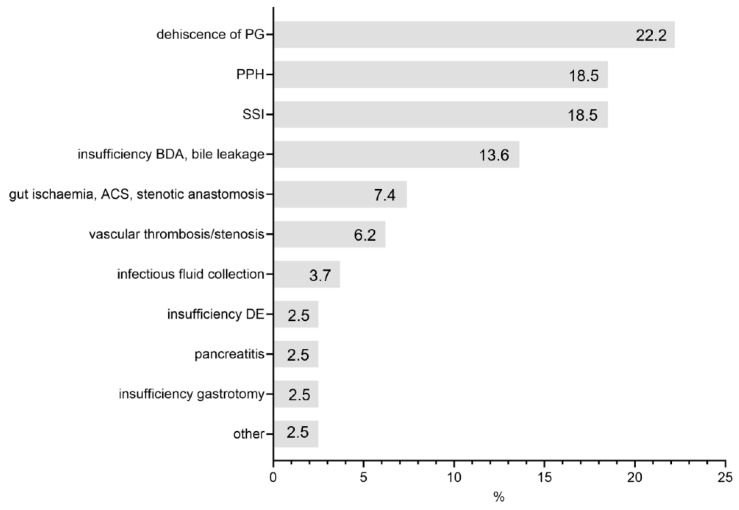
Indications for reoperation.

**Table 1 jcm-11-07512-t001:** Demographic and perioperative data of index operation.

	Operative re-Intervention	No Operative Re-Intervention	*p*
	n = 81	n = 328	
age (a)	67 (58–73)	66 (57–75)	0.956
age >80 years	4 (5%)	33 (10%)	0.150
gender female	24 (30%)	136 (41%)	0.051
BMI (kg/m^2^)	25.1 (22.3–28.7)	24.8 (22.5–27.7)	0.695
diagnosis malignant	58 (72%)	249 (76%)	0.422
alcohol abuse	27 (33%)	93 (28%)	0.273
nicotine (active consumption)	21 (26%)	98 (30%)	0.543
weight loss	36 (43%)	174 (50%)	0.117
Charlson Comorbidity Index	2 (1–3)	2 (2–3)	0.046
preoperative cholangitis	30 (30%)	53 (16%)	0.005
preoperative biliary drainage	34 (42%)	166 (51%)	0.193
previous operations	28 (35%)	142 (43%)	0.884

Data are shown as frequency (%) or median (interquartile range), BMI = body mass index.

**Table 2 jcm-11-07512-t002:** Perioperative data of index operation.

	Operative Re-Intervention	No Operative Re-Intervention	*p*
	n = 81	n = 328	
duration of operation (min)	458 (355–556)	401 (333–473)	0.003
blood loss (mL)	800 (475–1500)	600 (375–1000)	0.036
transfusions (erythrocyte concentrate)	1 (0–4)	0 (0–2)	0.001
positive intraoperative microbiology	21 (26%)	149 (45%)	0.075
tumor size (cm)	3.0 (2.0–4.0)	2.8 (2.0–3.8)	0.488
soft pancreas parenchyma	26 (32%)	96 (29%)	0.069
pancreatic duct >5 mm	9 (11%)	48 (15%)	0.815
extended lymphadenectomy	47 (58%)	211 (64%)	0.423
resected lymph nodes	25 (17–31)	22 (16–30)	0.199
venous reconstruction	13 (16%)	54 (16%)	0.920

Data are shown as frequency (%) or median (interquartile range).

**Table 3 jcm-11-07512-t003:** Postoperative outcome/complications of index operation.

	Operative Re-Intervention	No Operative Re-Intervention	*p*
	n = 81	n = 328	
Clavien major (grade III-IV)	75 (93%)	126 (38%)	<0.001
insufficiency of BDA	14 (17%)	11 (3%)	<0.001
insufficiency of DE	28 (35%)	4 (1%)	<0.001
wound infection (suprafascial)	39 (48%)	53 (16%)	<0.001
intraabdominal abscess formation	25 (31%)	35 (11%)	<0.001
PF grade B/C	36 (44%)	45 (14%)	<0.001
PPH grade B/C	30 (37%)	71 (22%)	0.002
DGE grade B/C	24 (30%)	63 (19%)	0.002
postoperative stay (d)	38 (22–64)	20 (15–26)	<0.001
stay in intensive care unit (d)	6 (2–16)	2 (1–3)	<0.001
stay in intensive care unit with respirator (d)	1 (0–7)	0 (0)	<0.001
mortality	20 (25%)	7 (2%)	<0.001

Data are shown as frequency (%), BDA = biliodigestive anastomosis, DE = duodenoenterostomy, PF = pancreatic fistula, PPH = postpancreatectomy hemorrhage, DGE = delayed gastric emptying.

**Table 4 jcm-11-07512-t004:** Risk factors associated with high mortality.

	Odds Ratio	95%-CI	*p*
**univariate**			
reoperation	15.035	6.093–37.099	≤0.001
PF grade B/C	3.045	1.354–6.846	0.005
PPH grade B/C	3.615	1.637–7.980	≤0.001
intraabdominal abscess formation	2.226	0.909–5.652	0.072
positive intraoperative microbiology	0.393	0.154–1.001	0.044
**multivariate**			
reoperation	15.394	5.794–40.897	≤0.001

BDA = biliodigestive anastomosis, DE = duodenoenterostomy, BMI = body mass index, CI = confidence interval.

**Table 5 jcm-11-07512-t005:** Comparison between 1989–2007 and 2008–2021 (Implementation ISGPS).

	1989–2007	2008–2021 ISGPS	*p*
	n = 285	n = 409	
age > 80 years	9 (3%)	37 (9%)	0.002
gender female	119 (42%)	160 (39%)	0.486
tumor size (cm)	2.65 (2.0–3.5)	2.90 (2.0–3.9)	0.436
duration of operation (min)	400 (341–461)	410 (335–460)	0.572
blood loss (mL)	800 (400–1300)	600 (400–1000)	0.075
transfusions (erythrocyte concentrate)	2 (0–4)	0 (0–2)	<0.001
PF yes/no	42 (15%)	160 (39%)	<0.001
PPH yes/no	31 (11%)	115 (28%)	<0.001
DGE yes/no	79 (28%)	176 (43%)	<0.001
reoperation	31 (11%)	81 (20%)	0.002
postoperative stay (d)	17 (14–25)	21 (15–30)	<0.001
stay in intensive care unit (d)	3 (1–4)	2 (1–4)	0.042
mortality	14 (5%)	27 (6%)	0.353

Data are shown as frequency (%), ISGPS = International Study Group of Pancreatic Surgery, PF = pancreatic fistula, PPH = postpancreatectomy hemorrhage, DGE = delayed gastric emptying.

## Data Availability

Our anonymized pancreatic resection database contains sensitive data (e.g., date of surgery), with which certain patients could be identified. According to German law and according to the approval of the ethics committee, these data must not be published. Access to the database can be obtained from the corresponding author upon reasonable request.

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
