# Peer review of "Operative Re-Intervention following Pancreatoduodenectomy: What Has Changed over the Last Decades"

_jcm, 2022, doi:10.3390/jcm11247512_

Round 1

Reviewer 1 Report

The authors present a very nice paper on operative re-intervention following pancreatoduodenectomy

Although they have performed similar research in the past, the findings in this paper are substantially different and of high interest, since pancreatic surgery is being more and more performed around the globe. The re-op is very important regarding the quality of life and mortality.

I would just stress in the discussion if there is a method of predicting the dehiscence. Recently some papers have addressed histological factors that are associated with this event and should add extra value to the manuscript. 

Author Response

First of all, we would like to thank the reviewer for her/his pro bono work and constructive remarks. We tried to meet the suggestions as follows:

Reviewer 1

The authors present a very nice paper on operative re-intervention following pancreatoduodenectomy

Although they have performed similar research in the past, the findings in this paper are substantially different and of high interest, since pancreatic surgery is being more and more performed around the globe. The re-op is very important regarding the quality of life and mortality.

Query 1:

I would just stress in the discussion if there is a method of predicting the dehiscence. Recently some papers have addressed histological factors that are associated with this event and should add extra value to the manuscript. 

Thank you very much for this valuable comment. Risk factors for pancreatic fistula, which is associated with dehiscence of pancreatogastrostomy, have been extensively studied. Our own group just recently analyzed risk factors, including histological factors, for pancreatic fistula that identified malignant histology and soft pancreatic parenchyma as risk factors for pancreatic fistula, whereas positive lymph node histology and pancraetic duct diameter <5mm were not associated with a higher risk for pancreatic fistula (data not shown, not published yet, under revision).

We added a paragraph to the discussion (page 9) in which we dealt with the various risk scores for the development of pancreatic fistula. We hereby emphasized histological factors that are associated with a higher rate of pancreatic fistula.

Reviewer 2 Report

1: An average mortality rate of 6% is mentioned in the introduction. However, the reference used for this statement analyzed only the effect of high-volume centers in Germany. Several internationally orientated papers report on lower mortality rates: 3% after PD. (Merath et al. J Gastrointest Surg. 2020, Smits et al, Ann Surg. 2022)

2: An important reference regarding the management of POPF is missing: Smits FJ et al. JAMA Surg. 2017. The authors from this article underpinthe fact that a minimal invasive approach (e.g., endoscopic or radiological intervention) is preferred over redo surgery for complications in pancreatic surgery. 

3: I would strongly advise to check the article by a native speaker. Grammer and punctuation could be improved to make the article more readable. For example line 60: accoording instead of according. Line 66: infilrtration... Line 72: peridual??

4: In the methods it is stated that for normally distributed variables the median and SD is given. For normally distributed variables one should present the mean and SD. Furthermore, in the results section none of the tables show variables with a mean given, so why is this stated in the methods section?

5: In the results section you use descriptive sentences: next, we wanted to identify.. This has already been (or should be) stated in the methods section. Solely present your results in the results section. 

6: preoperative cholangitis is a independent risk factor for postoperative mortality and therefore might be a confounder in your results: Darnell et al. Am. J. Surg. 2021

Author Response

First of all, we would like to thank the reviewer for her/his pro bono work and constructive remarks. We tried to meet the suggestions as follows:

Reviewer 2

Comments and Suggestions for Authors

Query 1:

An average mortality rate of 6% is mentioned in the introduction. However, the reference used for this statement analyzed only the effect of high-volume centers in Germany. Several internationally orientated papers report on lower mortality rates: 3% after PD. (Merath et al. J Gastrointest Surg. 2020, Smits et al, Ann Surg. 2022)

Nationwide reports and publications from centers of excellence for pancreatic surgery regularly differ in mortality. Furthermore, mortality rate after pancreatic resection differs geographically with lower mortality rates in the US (PMID: 33741182, PMID: 34132696).

For instance, the RECOPANC Study, a large controlled randomized German study, with only centers of excellence participating, reflected an in-hospital mortality rate of 6% in Germany (PMID: 26135690), and even in very high-volume centers with a mean of 133.8 pancreatic resections a year mortality is 6.1% (PMID: 28379871). Reasons for this higher mortality rate were debated and could be explained by different patient selection. We rewrote the section specifying that the mortality rate of 6% reflects the mortality rate in Germany, with even lower rates in the US (page 1).

Query 2:

An important reference regarding the management of POPF is missing: Smits FJ et al. JAMA Surg. 2017. The authors from this article underpin the fact that a minimal invasive approach (e.g., endoscopic or radiological intervention) is preferred over redo surgery for complications in pancreatic surgery. 

We agree, that treatment of POPF strongly depends on the degree of the fistula with an asymptomatic biochemical leak requiring no treatment and a symptomatic grade B/C POPF requiring prolonged drainage or additional percutaneous or transgastric drainage and grade C POPF being associated with organ failure and/or circulatory instability requiring intervention or reoperation. We already emphasized this in the discussion section on page 8  “Importantly, not every patient with dehiscence of PG required surgery. Only in case of additional extraluminal hemodynamic relevant bleedings and septic conditions a second surgery was required”. Additionally, we added the reference of Smits et al. (page 8).

Query 3:

I would strongly advise to check the article by a native speaker. Grammer and punctuation could be improved to make the article more readable. For example line 60: accoording instead of according. Line 66: infilrtration... Line 72: peridual??

We thoroughly revised the manuscript regarding grammar and punctuation with the help of a native speaker and highlighted the corrections throughout the manuscript. Specifically, we corrected the above mentioned spelling mistakes and changed the word peridual to epidural (page 2).

Query 4:

In the methods it is stated that for normally distributed variables the median and SD is given. For normally distributed variables one should present the mean and SD. Furthermore, in the results section none of the tables show variables with a mean given, so why is this stated in the methods section?

We sincerely apologize for this mistake. You are entirely right: for normally distributed variables one should present the mean ± SD. Our variables follow a non-normal distribution, which is why you cannot find variables with a mean in the tables. We corrected this mistake by deleting the sentence of normally distributed variables from the methods part (page 2) since our data follow a non-normal distribution.

Query 5:

In the results section you use descriptive sentences: next, we wanted to identify.. This has already been (or should be) stated in the methods section. Solely present your results in the results section. 

We changed this accordingly and removed the descriptive sentences as suggested and added the information to the methods section (page 2).

Query 6:

preoperative cholangitis is a independent risk factor for postoperative mortality and therefore might be a confounder in your results: Darnell et al. Am. J. Surg. 2021

We discussed this in the discussion section (page 7): “Preoperative cholangitis was found to be an independent risk factor for mortality after PD (PMID: 32847686). Since our patients in the reoperation group showed more frequently preoperative cholangitis, this could account for the high mortality observed in this group - at least to some extent. However, the results have to be interpreted carefully and cannot be transferred to our study in full extent since the study by Darnell et al. only included patients that pursued neoadjuvant chemotherapy followed by resection. In our study only 16 patients, out of the 409 patients included, received chemotherapy before surgery. Interestingly, preoperative cholangitis did not have an impact on postoperative complications in the mentioned study”.